# *Olea europea* L. Leaves and *Hibiscus sabdariffa* L. Petals Extracts: Herbal Mix from Cardiovascular Network Target to Gut Motility Dysfunction Application

**DOI:** 10.3390/nu14030463

**Published:** 2022-01-21

**Authors:** Laura Beatrice Mattioli, Maria Frosini, Rosa Amoroso, Cristina Maccallini, Elda Chiano, Rita Aldini, Francesco Urso, Ivan Corazza, Matteo Micucci, Roberta Budriesi

**Affiliations:** 1Department of Pharmacy and Biotechnology, Food Chemistry and Nutraceutical Lab, Alma Mater Studiorum-University of Bologna, 40126 Bologna, Italy; laura.mattioli13@unibo.it (L.B.M.); rita.aldini@unibo.it (R.A.); francesco.urso95@gmail.com (F.U.); 2Department of Life Sciences, University of Siena, 53100 Siena, Italy; maria.frosini@unisi.it (M.F.); elda.chiano@unisi.it (E.C.); 3Department of Pharmacy, University “G. d’Annunzio” of Chieti-Pescara, 66100 Chieti, Italy; rosa.amoroso@unich.it (R.A.); cristina.maccallini@unich.it (C.M.); 4Department of Experimental, Diagnostic and Specialty Medicine-DIMES, Alma Mater Studiorum-University of Bologna, 40138 Bologna, Italy; ivan.corazza@unibo.it; 5UniCamillus-Saint Camillus International University of Health Sciences, Via di Sant’Alessandro, 800131 Rome, Italy; matteo.micucci@uniurb.it; 6Department of Biomolecular Sciences, Università degli Studi di Urbino “Carlo Bo”, 61029 Urbino, Italy

**Keywords:** *Olea europea* L. leaves extract, *Hibiscus sabdariffa* L. calyces extract, ileum and colon contractility, human iNOS (target) and eNOS, human colorectal adenocarcinoma

## Abstract

It is well known that diet and nutrition play a critical role in the etiopathogenesis of many disorders. On the other hand, nutrients or bioactive compounds can specifically target and control various aspects of the mechanism underlying the pathology itself, and, in this context, diseases related to intestinal motility disorders stand out. The Herbal Mix (HM) consisting of *Olea europea* L. leaf (OEE) and *Hibiscus sabdariffa* L. (HSE) extracts (13:2) has been proven to be a promising nutraceutical option for many diseases, but its potential in inflammatory-driven gastrointestinal disorders is still unexplored. In this study, HM effects on guinea-pig ileum and colon contractility (induced or spontaneous) and on human iNOS activity, as well as on human colorectal adenocarcinoma Caco-2 cells, were studied. Results showed that the HM can control the ileum and colon contractility without blocking the progression of the food bolus, can selectively inhibit iNOS and possesses a strong pro-apoptotic activity towards Caco-2 cells. In conclusion, the present results suggest that, in some diseases, such as those related to motility disorders, an appropriate nutritional approach can be accompanied by a correct use of nutraceuticals that could help not only in ameliorating the symptoms but also in preventing more severe, cancer-related conditions.

## 1. Introduction

Phytocomplexes are pools of molecules contained in plants that often have an interesting biological value that justifies their use as an ingredient in food supplements or drug formulations. Their potential lies in the fact that these organic compounds bind to the same targets as drugs. In most cases, these are functional analogues without structural similarity among themselves, which makes it more complicated to identify the mechanisms underlying their effects and to represent the link between food and drugs [1]. Moreover, the biological activity of the phytocomplex does not correspond to that of the isolated compounds, as multiple synergies and antagonisms among its components, as well as between them and the specific molecular targets, occur, thus contributing to its biological activity as “a whole”.

In recent years, our research group has adopted the target network strategy (i.e., multi-drugs → multi-targets → multi-diseases) to demonstrate and justify the beneficial use of phytocomplexes for treating specific pathologies, owing to the fact that a phytocomplex might act on multiple targets rather than selectively bind to one of them [2,3,4,5,6]. Broadening the characterization of their activity, we have shown that the interaction with different networks that are “off target” for the pathology under study can be of great interest for treating other diseases. These networks and the relative interconnections, in fact, provide useful information for finding new targets, and, as a consequence, new drugs. In addition to this, there is the consolidated approach, adopted also for many food supplements, of preparing mixed extracts with a dual purpose: to broaden the target network and to reduce any side effects, in agreement with what is performed with the use of herbal mix in the traditional Chinese medicine [7].

Our research group recently developed an Herbal Mix (HM) consisting of *Olea europea* L. leaves extract and *Hibiscus sabdariffa* L. calyces extract in a ratio of 13:2, respectively, that is potentially useful for treating cardiovascular diseases [8]. The resulting formulation was proven to be a promising nutraceutical option for the management of the initial stages of hypertension, in that the HM mainly acts as an L-type calcium channel (LTCC) blocker [9]. It is well known, however, that these channels also play a key role outside the cardiovascular system, such as in the brain and gut [10,11,12], further highlighting their great therapeutic potential [13]. The preliminary results on ileum, however, suggested that the HM possesses myorelaxant activity accompanied by changes in the contraction frequency, which may result in slowing gut transit as it occurs in hypertensive patients treated with calcium antagonists [9]. This encouraged us to deepen its effects in a reversed perspective: in the present study, HM effects, along with those of its components, on ileum and colon (target) were assessed, comparing them with those at the cardiovascular level (off-target). Moreover, keeping in mind the “network target-based” approach, we have extended the investigation to other targets with the aim to verify whether the HM can be useful for treating/preventing gastrointestinal disorders, such as inflammatory bowel disease (IBD) and colorectal cancer (CRC). As the initiation and maintenance of inflammation in IBD has been associated with the up-regulation of the production of nitric oxide (NO) via the expression of the inducible isoform of the NO synthase (iNOS) [14], iNOS inhibitors could represent a therapeutic strategy for IBD, as well as all those conditions in which NO overproduction plays a pathological role [15,16]. Moreover, the overexpressed iNOS and inflammation are also associated with CRC [17], and this is supported by the fact that patients with IBD (ulcerative colitis or Crohn’s disease) have a higher risk of developing CRC, the risk being higher in the case of chronic inflammation [18]. For these reasons, the effects of the HM, along with those of its single components on human iNOS (target) and eNOS (off-target), as well as on human colorectal adenocarcinoma Caco-2 cell viability (target), were assessed in order to highlight the HM potential use in inflammatory-driven gastrointestinal disorders.

## 2. Materials and Methods

### 2.1. Plant Materials

The *Olea europea* L. leaves extract Benolea EFLA 943^®^ (OEE) was supplied by Frutarom Switzerland Ltd. (Wädenswil, Switzerland) and prepared as previously described [19]. The *Hibiscus sabdariffa* L. flowers extract (HSE) provided by Nutraceutica s.r.l (Monterenzio, Bologna, Italy) was prepared as previously reported [8]. More information on the extracts’ preparation and their chemical characterization is available [20]. The herbal mix (**HM**) was formed by 86.67% of OEE and 13.33% of HSE (13:2 ratio).

### 2.2. “Ex Vitro” Studies

Male guinea-pigs (200–400 g) (Charles River, Calco, Como, Italy), were housed according to the ECC Council Directive concerning use of animals for scientific research. All experiments were conducted according to EU Directive 2010/63/EU and to ARRIVE guidelines [21] and approved by the Institutional Ethics Committee of the University of Bologna (Protocol PR 21.79.14).

#### 2.2.1. Ileum and Colon Contractility

Ileum. The terminal portion of ileum (3–4 cm near the ileo-caecal junction) was cleaned, and cylindrical segments 1–2 cm long were set up under 1 g tension at 37 °C in 15 mL organ baths containing Tyrode solution (composition in mM: NaCl, 118; KCl, 4.75; CaCl_2_, 2.54; MgSO_4_·7H_2_O, 1.20; KH_2_PO_4_·2H_2_O, 1.19; NaHCO_3_ 25; glucose 11) consistently warmed (see below) and buffered to pH 7.4 by saturation with 95% O_2_–5% CO_2_ gas. The segments were set up under 1 g tension in the longitudinal direction along the intestinal wall and allowed to equilibrate for at least 30 min, during which, the bathing solution was changed every 10 min.

Colon. Starting approximately 1 cm distal from the caecocolonic junction, two cylindrical segments of approximately 1 cm of the guinea-pig proximal colon were cut and cleaned by using De Jalon solution (composition in mM: NaCl, 155; KCl, 5.6; CaCl_2_, 0.5; NaHCO_3_, 6.0; glucose, 2.8); afterward, the mesenteric tissue was removed. The two segments were suspended in organ baths containing gassed (95% O_2_–5% CO_2_ gas) warm de Jalon solution under a load of 1 g maintained at 37 °C. Tension changes in longitudinal muscle length were recorded. Tissues were allowed to equilibrate for at least 30 min, during which time, the bathing solution was changed every 10 min.

Induced. L-type calcium channel (LTCC)-mediated spasmolytic activity was studied using ileum and colon longitudinal smooth muscle contracted by high K^+^ concentration (80 mM). Tension changes in smooth muscle relaxation were recorded isometrically as previously described [4].

Spontaneous. The LabChart Software (ADinstruments, Bella Vista, NSW, Australia) was used to acquire the tracing graphs of spontaneous contractions of ileum and colon. After the equilibration period (approximately 30 to 45 min according to each tissue), cumulative concentration curves (0.1, 0.5, 1.0, 5.0 and 10.0 mg/mL) of HM, OEE and HSE samples were constructed. Cumulative log-dose–response curves were constructed for Otylonium Bromide (OB), used as a positive control. A 5 min stationary interval of the Spontaneous Contraction (SC) recording was selected at the final stage of each concentration. For each interval, the following parameters were extracted and calculated: the Mean Contraction Amplitude (MCA), evaluated as the mean force value (g); the standard deviations of the force values over the period, as an index of the Spontaneous Contraction Variability (SCV); and Basal Spontaneous Motor Activity (BSMA), as the percentage (%) variation of each mean force value (g) with respect to the control period. The spontaneous contractions were investigated in the frequency domain through a standard FFT analysis and a subsequent Power Spectral Density (PSD) plot. The absolute powers of the following frequency bands of interest—low [0.0,0.2[ Hz (LF), medium [0.2,0.6[ Hz (MF) and high [0.6,1.0] Hz (HF) [5]—were then calculated. The PSD percentage (%) variations for each band of interest with respect to control were estimated.

All of the calculations were carried out in a post-processing phase. To avoid errors due to the presence of artifacts, the period of analysis was chosen by a skilled operator.

#### 2.2.2. Atria

After the explant, heart was washed by perfusion through the aorta with oxygenated Tyrode solution. Spontaneously beating right and left atria driven at 1 Hz were used. The contractile activity on right and left atria was recorded isometrically. All effects are recorded using cumulative concentration curves (0.1, 0.5, 1.0, 5.0 and 10.0 mg/mL) of HM, OEE and HSE samples that were constructed. Cumulative log-dose–response curves were constructed for OB and used to compare the off target activity [9].

#### 2.2.3. Aorta

As previously described [8], the thoracic aorta was removed and placed in Tyrode solution containing (mM): NaCl, 118; KCl 4.75; CaCl_2_ 2.54; MgSO_4_ 1.20; KH_2_PO_4_ 1.19; NaHCO_3_ 25; and glucose 11; bubbled with 95% O_2_–5% CO_2,_ pH 7.4. The vessel was cleaned of extraneous connective tissue. After the equilibration period, guinea-pig aortic strips were contracted by washing in PSS containing 80 mM KCl (equimolar substitution of K^+^ for Na^+^). When the contraction reached a plateau (approximately 45 min or 15 min, respectively) HM, OEE and HSE samples were added cumulatively to the bath, allowing for any relaxation to obtain an equilibrated level of force. Cumulative concentration curves (0.1, 0.5, 1.0, 5.0 and 10.0 mg/mL) were constructed. Cumulative log-dose–response curves were constructed for OB and used to compare the off target activity.

### 2.3. NOS Assay

#### 2.3.1. General

For the evaluation of NOS inhibition, the L-citrulline assay with fluorimetric detection was followed as previously described [22]. HPLC analyses were performed using a Waters (Milford, MA, USA) system composed of a P600 model pump, a 2996 photodiode array detector and a 7725i model sample injector (Rheodyne, Cotati, CA, USA). Chromatograms were recorded on a Fujitsu Siemens Esprimo computer and the Empower Pro software (Waters) processed data. The analyses were performed on an XTerra MS C8 column (250 × 4.6 mm id, 5 μm particle size) (Waters), equipped with an XTerra MS C8 guard column (Waters). A column thermostat oven module Igloo-Cil (Cil Cluzeau Info Labo, Sainte-Foy-la-Grande, France) was used.

#### 2.3.2. Procedure

Recombinant human iNOS was purchased from Enzo Life Sciences, Inc. (New York, NY, USA). Recombinant bovine eNOS was purchased from Cayman Chemical (Ann Arbor, MI, USA). To measure the NOS activity, 10 μL of the test extract solution (100–1 mg/mL) or of the positive control L-NAME (1 mM) was added to the enzyme assay solution [22], followed by pre-incubation of 15 min at 37 °C. Then, 10 μL of nicotinamide adenine dinucleotide phosphate (NADPH) 7.5 Mm was added to each reaction, and, after 30 min at 37 °C, 500 μL of ice-cold CH_3_CN were added. The mixture was dried under vacuum and then subjected to the L-citrulline fluorescence derivatization [22]. The content of the L-citrulline was evaluated with respect to the NOS reaction control, where no extract nor inhibitor was added (0% inhibition), whereas L-NAME was the positive control (100% inhibition).

### 2.4. Human Colorectal Adenocarcinoma Caco-2 Cells Assays

Human colorectal adenocarcinoma Caco-2 cells (ATCC^®^ HTB-37™, passages 10–20) were grown in standard conditions as previously reported [23]. HM solutions were prepared immediately before use by appropriate dilution of 10.0 mg/mL stock solution. Caco-2 cells (1 × 10^4^ cells/well) were incubated with the extract (0–500 µg/mL, 24 h) and, afterward, MTT assay was performed [24]. Changes in cell viability were also checked by a phase-contrast light microscope, and the resulting images were analyzed by blinded operator using the grade scale as already reported [25]. Reversibility of HM-mediated cytotoxicity was also tested. Briefly, Caco-2 cells were incubated for 24 h with HM. Afterward, the medium was removed and replaced by fresh HM-free culture medium, and cells were incubated for additional 24 or 48 h, assessing their viability at each time point [24].

#### 2.4.1. Apoptosis and Mitochondria Integrity Assays

Cell cycle and sub-G0/G1 population analysis, annexin V/propidium iodide (AV/PI) or rhodamine-123 (R123) staining were used for checking apoptosis and mitochondria integrity by using flow cytometry. A FACScan flow cytometer equipped with CellQuest software v. 3.0 (BD Biosciences, San Jose, CA, USA) was used, and the fluorescence of 10^4^ single cells/sample was acquired. Samples were excited at 488 nm and the emission fluorescence was detected at 530 ± 30 nm (FL1, PI or R123) or 585 ± 42 nm (FL2, AV) [26].

#### 2.4.2. Ileum or Colon Rings

The effects of HM on a more complex and physiological context, such as the whole healthy tissue, were also investigated in rat ileum and colon rings. These were prepared and treated with HM according to the protocol already described [24]. Viability of tissue was assessed immediately after ring preparation (fresh tissue) or after 24 h incubation with HM (0–1000 µg/mL) by MTT assay [24].

### 2.5. Data Analysis

The results were expressed as mean ± SEMs or SD. Statistical significance was assessed by using ANOVA followed by Dunnett or Bonferroni post-test or Student’s *t*-test, as appropriate (GraphPad Prism version 5.04, GraphPad Software Inc., San Diego, CA, USA). In all comparisons, the level of statistical significance (*p*) was set at 0.05.

The spasmolytic activity was evaluated on calcium-induced contractility guinea-pig ileum and colon strips or rings (smooth muscle activity) (*n* = 6–8), and expressed as the percentage inhibition of maximal contraction elicited. The potency of OEE, HSE, HM and OB (positive control) was defined as EC_50_ or IC_50,_ which was calculated from the concentration- or log-concentration vs. response curves (Probit analysis using Litchfield and Wilcoxon [27] or GraphPad Prism (version 5.04, GraphPad Software Inc., San Diego, CA, USA) [28]. In guinea-pig spontaneous contractility (*n* = 6–8), samples were added in a cumulative manner and the difference between the control and the experimental values at each concentration was tested for statistical significance. On spontaneous contractility experiments, data from concentration–response curves were analyzed as described above [28,29].

iNOS IC_50_ values were calculated by means of the GraphPad Prism version 5.04 (GraphPad Software Inc., San Diego, CA, USA).

## 3. Results

### 3.1. Chemistry

Table 1 reports the chemical composition of the photocomplexes used to prepare the HM determined by HPLC-MS/MS analysis. As evidenced, HSE mainly consists of hibiscus acid (Figure 1), whereas, in OEE, secoiridoids consist mainly of oleuropein (Figure 1) and its isomers. As a result, with the HM being formed by 86.67% of OEE and 13.33% of HSE, its main components are secoiridoids, which consist mainly of oleuropein and its isomers, followed by hibiscus acid (Figure 1).

### 3.2. Target: In Vitro Ileum and Colon Contractility

#### 3.2.1. Induced Contractility on K^+^ (80 mM)-Depolarized Tissues

OEE, HSE and HM have been studied for their spasmolytic effects on the longitudinal smooth muscle of ileum and depolarized colon K^+^ (80 mM) in order to study their effects on LTCC. The results are summarized in Table 2.

The OEE and HM showed an interesting spasmolytic activity that reaches 90% at 10.0 mg/mL, with a potency of 0.77 mg/mL, whereas HSE was mostly ineffective. It is interesting to underline that, on ileum, the HM has an intrinsic activity that is not significantly different from OEE, even though the potency (1.39 mg/mL) is approximately 1.8 times lower as previously described [9]. The same trend is also evident on the colon but at lower concentrations: OEE is more active and more powerful than the HM with a similar selectivity.

HSE has no significant action on the colon either. Even OB, taken as a reference compound, has a greater intrinsic activity and potency on the colon than ileum, with a selectivity of approximately 2.5 times. OEE and HM have a colon selectivity of around 5.

#### 3.2.2. Spontaneous Contractility

OEE, HSE and HM effects have also been studied on the spontaneous contractility of both longitudinal and circular ileum and colon smooth muscle. All of the experiments were also conducted with OB taken as the positive control.

Ileum smooth muscle: OEE (Figure 2b) shows a positive trend of tone increase up to 5.0 mg/mL. At 10.0 mg/mL, this parameter seems to return to control values; thus, between 0.1 and 10.0 mg/mL, there is not a significative modification. High and medium frequency contractions underwent a significant decrease. Concerning circular contractions, no tone modifications were observed (Figure 3b); however, low and medium frequency contractions weakly increased at 0.1 mg/mL, and slightly and progressively decreased at the following concentrations (Figure 3c). 

HSE treatment did not produce any modifications of tone; however, there was a decrease only at high concentrations (10.0 mg/mL) (Figure 2b). The PSD plot (Figure 2c) shows a decrease in all the bands at 10.0 mg/mL. Concerning circular contractions, HSE produces a reduction in tone with concentration (maximal effect at 5.0 mg/mL) (Figure 3b), whereas the motility at low medium frequencies increases only at 5.0 mg/mL (Figure 3c). 

The HM slightly increases the ileum smooth muscle tone up to 0.5 mg/mL, and then it decreases it up to 10.0 mg/mL (Figure 2b), whereas it determines a decrease in motility at all frequencies from 1 to 10.0 mg/mL (Figure 2c).

Concerning circular smooth muscle contractility (Figure 3), the HM has a weak effect (reduction) on the tone (Figure 3b), while all the frequency bands decrease with an increase in concentration.

Upon the OB treatment of longitudinal muscle (Figure 4b), the tone undergoes a progressive decrease up to a value of 60%. High and medium frequency contractions undergo a reduction (Figure 4c), likely resulting in antinociceptive activity, whereas low frequencies are not significantly affected, with a weak decrease at 10 µM.

The circular smooth muscle contractility is not substantially altered (Figure 5).

The results relating to the spontaneous contractility of the ileum are summarized in Table 3.

Colon smooth muscle. On longitudinal contractions, OEE determines a tone decrease up to 5.0 mg/mL, followed by an increase with values even lower than those of the control (Figure 6b). Low and medium frequency contractions undergo an increase of up to 0.5 mg/mL and a strong decrease at 5.0 and 10.0 mg/mL (Figure 6c).

HSE does not produce a significant tone reduction, except for 1.0 mg/mL (Figure 6b). Low frequency contractions increase up to 1.0 mg/mL, and decrease at 5.0 and 10.0 mg/mL. Medium frequency contractions increase up to 0.5 mg/mL and undergo a powerful reduction at 1.0, 5.0 and 10.0 mg/mL (Figure 6c).

The HM determines a strong tone decrease in a concentration-dependent manner (Figure 6a,b). In addition, the amplitude decreases in a concentration-dependent manner. The HM induces a reduction in all frequencies’ contractions (Figure 6).

OB induces a decrease in the tone at 10 µM (Figure 7b). At the highest concentration, the high frequency contractions also undergo a significant reduction (Figure 7c).

On the colon circular smooth muscle, OEE seems to produce a weak decrease in the tone of the muscle contractility (Figure 8b).

HSE reduces the tone up to 1.0 mg/mL, whereas it increases it at 5.0 and 10.0 mg/mL (Figure 8b). This extract increases low and medium frequency contractions at 0.5 and 1.0 mg/mL, whereas it induces a strong reduction in these frequencies’ contractions at 10.0 mg/mL (Figure 8c). 

HM reduces the tone in a concentration-dependent manner (Figure 8b). In addition, a slight increase in the low frequency contractions occurs along with the increasing concentration, whereas a medium frequency increase is observed at 10.0 mg/mL (Figure 8c).

Regarding colon circular smooth muscle, OB does not induce any tone modifications (Figure 9b). Neither MCA nor BSMA undergo any alterations. OB determines a slight decrease in the low and medium frequency contractions (Figure 9c) at the highest concentration tested (10 µM).

The results, relating to the colon spontaneous contractility, are summarized in Table 4.

#### 3.2.3. Off Target: Effect on Cardiovascular System

OB is a calcium antagonist used in the therapy for motility disorders [13]. Owing to its mechanism of action, its effects on the cardiovascular parameters, which represent an off target, were assessed. In Table 5, previous [9] and new data describing the effects of OEE, HSE and HM on the left and right atrium, as well as on aorta contractility, are reported and compared with those elicited by OB, which is taken as a positive control.

The HM has selectivity on the ileum and colon four and two times, respectively, when compared with vasorelaxant potency (Table 5). OB is inactive on vascular smooth muscle. As for the negative chronotropic effects, the HM has a similar action on the ileum but is approximately five times less potent than the colon. Like OB, the HM induces negative inotropic activity on our experimental models. However, OB is 228 and 90 times more potent towards negative inotropy than spasmolytic potency on the ileum and colon, respectively, whereas the HM has the same potency on ileum and is only eight times more potent on negative inotropy than on the colon.

### 3.3. Target and off Target: iNOS and eNOS

The potential inhibition of the iNOS (target) by the HM was evaluated and compared with the effects on the constitutive endothelial NOS (eNOS) (off target) to assess the extract selectivity with respect to this isoform, which plays essential roles in the cardiovascular system.

Firstly, both of the pure HM ingredients, i.e., OEE and HSE, were incubated with the NOS at the concentration range 0.1–10.0 mg/mL. In Figure 10a, the calculated iNOS and eNOS percentage inhibition values at each OEE assayed concentrations are reported. As for the iNOS inhibition, the extract was able to inhibit the enzyme in a dose-dependent manner, with an inhibition range of 24–86% and an iNOS IC_50_ of 0.485 mg/mL (Table 6). Similarly, the eNOS was also inhibited, with a maximal effect at the highest doses of 5.0 and 10.0 mg/mL (97% and 98% inhibition, respectively), with an IC_50_ of 1.460 mg/mL. A residual eNOS inhibition of 12% was evaluated at 0.5 mg/mL, while a complete isoform selectivity (0% inhibition) was reached at the lowest dose of 0.1 mg/mL (Table 6).

As for the effects of the HSE on iNOS and eNOS, no inhibition was observed. 

As for the effects of the HSE on iNOS and eNOS, no inhibition was observed. Finally, the two-extracts-based formulation (HM) was incubated with both the NOSs, and the results are reported in Figure 10b. A dose-dependent inhibition of both the two enzymes was observed, with a maximum effect of 67% for the iNOS and 64% for the eNOS at 10.0 mg/mL. The iNOS inhibition was also found at the lowest dose of 0.1 mg/mL (11%), whereas eNOS resulted in not being inhibited by the HM at 0.1 and 0.5 mg/mL. The extrapolated HM IC_50_ was 0.743 mg/mL for the iNOS and 2.051 mg/mL for the eNOS (Table 6), with a selectivity ratio eNOS/iNOS of 2.760.

### 3.4. Additional Target: Anticancer Activities

The potential of the HM toward CRC was assessed by treating human colorectal adenocarcinoma Caco-2 cells with an increasing concentration of the HM for 24 h and performing a MTT assay afterward. The results showed that the Caco-2 viability did not change up to 100 µg/mL, whereas it was significantly reduced upon HM 250 µg/mL and higher concentrations. The cytotoxic effect was concentration-dependent, with an IC_50_ value of 264.6 ± 10.2 µg/mL (Figure 11a). The observation of treated cells at contrast phase microscopy confirmed the MTT data, as HM-treated cells exhibited considerable changes in both their size and number and resembled an apoptotic cell’s typical appearance. These differences were increasingly evident at 250 µg/mL and 500 µg/mL, corresponding to cytotoxicity grades 2 and 3–4, respectively (see Figure 11b).

To verify whether the HM cytotoxic effect was reversible or not, Caco-2 cells were initially treated with the HM for 24 h, followed by adding a fresh HM-free culture medium and assessing the cell viability after a further 24 or 48 h of incubation. 

250 and 500 µg/mL of the HM, which caused a grade 2 or 3 of toxicity, was tested, and results showed that it caused an irreversible cytotoxic effect at both concentrations, as highlighted by the severe fall in the viability (Figure 11c), which was still clear even after an additional 48 h period of incubation.

As previously mentioned, the HM is a 13:2 (*w*/*w*) mix of the extracts prepared from *Olea europea* L. leaves and *Hibiscus sabdariffa* L. calyces, respectively. Thus, we found it of interest to compare the observed overall effect of the HM with those elicited by *Olea europea* L. (OEE) and *Hibiscus sabdariffa* L. (HSE) components by themselves. The effects of the corresponding amount found in 250 µg (217.0 µg of OEE and 33.0 µg of HSE) and 500 µg (434.0 µg of OEE and 66.0 µg of HSE) were thus singularly tested along with the resultant mixture (OEE + HSE, corresponding to HM) on the Caco-2 cell viability (Figure 11d). Interestingly, (OEE + HSE)-mediated cytotoxicity was always significantly higher than that caused by the single components (OEE or HSE). In particular, the latter were mostly ineffective when assessed singularly at 267.0 µg and 33.0 µg, respectively, whereas (OEE + HSE) 250 µg halved the number of viable cells, thus suggesting a synergistic effect. At the higher concentration, 434 µg of OEE as well as 66 µg of HSE reduced the cell viability by ~57.0 % and ~28.0%, respectively, whereas the viability detected after 500 µg of (OEE + HSE) was still significantly lower (~13.0 %, *p* < 0.05 vs. OEE and *p* < 0.001 vs. HSE). 

Finally, the effects of the HM on healthy tissue were also assessed in the rat proximal colon and ileum rings. The results demonstrated that the HM (up to 1000 µg/mL, 24 h) did not affect ilia or proximal colon rings’ viability, suggesting a safe profile of the extract (Figure 11e). 

To further study the mechanisms causing cell death, HM concentrations of 250 and 500 µg/mL (grade 2 or 3 of toxicity), as well as 100 µg/mL (the highest HM safe concentration), were used to perform some apoptosis assays. 

As reported in Figure 12a, the challenge with the HM caused a clear, concentration-dependent increase in early apoptotic cells, which was accompanied by a gradual reduction in healthy cells (AV- and PI-negative). Late apoptotic cells were not affected, whereas necrotic cells rose significantly only at the highest HM concentration of 500 µg/mL. The cell cycle analysis showed that the HM caused a concentration-dependent increase in sub-G0/G1 hypodiploid cells, which was paralleled by a reduction in those in the G0/G1 phase, with 500 µg/mL of HM endowed with the most striking effect (Figure 12b).

Caco-2 cells in the s phase were not changed at all by the treatment, whereas those in G2/m decreased only at the highest HM concentration, although not significantly. Changes in the mitochondria membrane potential (MMP), one of the early events in apoptosis, was also assessed. The percentage of cells with a loss in MMP increased significantly for both 250 and 500 µg/mL HM concentrations (Figure 12C), suggesting the involvement of mitochondria in HM-induced apoptosis.

## 4. Discussion

In functional intestinal disorders such as IBS, many factors contribute to the etiology of the disease. These include the change in intestinal motility, inflammation and the increase in intestinal permeability [30,31] resulting in an increased visceral sensitivity. For this reason, visceral hypersensitivity is considered as a crucial node of the network target linked to gastrointestinal disorders. The literature reports that visceral hypersensitivity plays a key role in the development and persistence of symptoms in most patients with IBS [32].

The gut smooth muscle cells contractility is regulated by a wide range of mechanisms, ranging from the release of neurotransmitters (such as acetylcholine, noradrenaline, histamine, etc.) and hormones such as prostaglandins. For many of them, the effect is obtained through an increase in the intracellular calcium concentration. Among all of the mechanisms potentially involved in contraction used as targets for motility control, LTCC are one of the most important. In fact, gut-selective LTCC are used in therapy for the control of functional motor disorders of the gut, such as in IBD. In addition, the blocking of LTCC results in reduced activity of all receptors activated by them, such as muscarinic receptors [2]. Among them, the postsynaptic M_3_ receptors, in fact, are involved in the contraction of intestinal smooth muscle and are over-expressed in pathologies such as diverticular disease, thus contributing to visceral hypersensitivity [33] and representing a possible target in the clinical application for this pathology [34]. Unfortunately, the lack of selectivity of many LTCC limits their use for gastrointestinal disorders, unlike **OB**, which is used in clinical practice to counteract abdominal pain in IBS mainly due to its poor oral absorption [13,35].

The ability of oleuropein, isolated from *Olea europea* L. leaves, to inhibit the tone of spontaneous activity in small rabbit intestine isolated segments was already described in 1972. Oleuropein also inhibits the BaCl_2_-induced contraction of isolated guinea-pig ileum, acetylcholine, histamine and nicotine receptors on different intestine segments [36]. Moreover, oleuropein exerts an interesting nociceptive effect through the blocking of LTCCs [37] and has well-known anti-inflammatory properties [38].

As highlighted in the results, OEE blocks calcium channels in both ileum and colon tissues. In the colon, it has a higher potency. HSE has no significant effect on this target. As evidenced by the results, the HM maintains the same activity profile as OEE. However, it is interesting to point out that the HM’s action trend is similar to that of OB [13,39]. The comparison with OB in quantitative terms is obviously difficult, as it is a drug with a high intrinsic activity and potency. However, it is scarcely selective because, even as demonstrated in our experimental models, OB has a significant action at the cardiac level (Table 5) on the inotropy. Interestingly, the HM has effects on the heart, but these are mild, making it better. Results, however, showed that the HM acts on the ileum and colon at concentrations close to those affecting the cardiovascular parameters. Despite these effects being assessed in ex vivo experiments, the possibility that this poor selectivity also occurs in vivo cannot be ruled out. If this is the case, as far as bioavailability and bio-accessibility is concerned, it will be necessary to set up a nutraceutical formulation whose composition drives the active compounds towards the tissue of interest. In other words, the formulation, as well as the excipients for targeting the cardiovascular system, will be different from that used for gastro-intestinal diseases [40].

We extended the characterization to the spontaneous contractility of ileum and colon longitudinal and circular smooth musculature for both OB and the HM.

Regarding the spontaneous contractility of ileum smooth muscle, OB has significant action on MCA ± SCV and on BSMA, resulting in a dose-dependent increase in intestinal lumen. At the same time, it does not significantly modify the peristalsis waves, thus suggesting that it maintains the progression of the bolus. The HM acts similarly on the same parameters in analogy with OEE. The effects of OEE on the wave panel are attenuated in the HM by the presence of HSE, leading to a dose-dependent effect. OB has no significant effect on colon longitudinal smooth muscle. The profile for HM is similar. Furthermore, in this case, the combination in the HM of OEE with HSE leads to a dose–effect relationship with respect to OEE, as shown in Figure 8. On the circular component of the ileum smooth musculature, OB has no significant effects on any of the monitored parameters, proved by the effect shown in the clinical application: a reduction in spasms. The HM has the same profile. In particular, the contractile effects of HSE are nullified by the combination with OEE.

Using what has been stated, we can hypothesize that the HM produces effects on spontaneous contractility in line with those produced by the reference drug OB: it allows us to control contractility without blocking the progression of the food bolus. 

The anti-inflammatory and antidiarrheal effects of HSE have already been documented [41]. In a high fat diet mice model, HSE reduced proinflammatory mediators and demonstrated prebiotic effects [42]. Similar effects on the microbiota were found in an in vitro simulation model of the human colon [43]. The same anti-inflammatory effects were seen for *Hibiscus rosa sinensis* Linn. leaf extract in an experimental animal model of colitis [44]. Interestingly, this extract inhibits some pro-inflammatory mediators, such as NO. The present results indicate a clear similarity between the HM and OB on the ileum and colon. Both have cardiovascular effects, but, whereas for OB they are particularly pronounced, for HM they are milder. In order to reduce/avoid HM effects at a cardiovascular level, a possibility could consist in the set-up of an appropriate tissue specific formulation that is able to selectively accumulate active compounds in the target organ (intestine) while keeping their concentration in off target organs (heart, blood vessels) below those that trigger cardiovascular effects. Interestingly, colon-specific drug delivery systems have been recently found, and these have had a positive significant impact for the treatment of intestinal inflammation-related diseases [40,45,46,47]. 

In addition, the HM acts on the network target by combining significant antitumor and anti-inflammatory effects linked to gastrointestinal disorders of ileum and colon contractility.

Oleuropein, a secoiridoid present in OEE, shows an anti-inflammatory effect acting on COX 2 in a model of murine colitis [48]. It reduces the production of NO, IL 1β, IL6 and TNFα, and the expression of iNOS and COX2: these effects are also observed in colon cell cultures stimulated with LPS [48]. The anti-inflammatory effect of oleuropein (500 mg/kg/day) was also confirmed in a mouse model of chronic colitis induced by dextran sodium sulphate (DSS) [49]. Studies carried out on animal models of IBD have shown that olive leave extract also attenuates chronic intestinal inflammation through the inhibition of iNOS, increasing the protective effect [50]. As iNOS inhibition represents a node of the target network linked to intestinal inflammatory pathologies and owing to the effect of oleuropein, we investigated the effects of OEE, HSE and the HM on iNOS. In addition, we studied the effects on eNOS, since its inhibition should be prevented to avoid side effects. Results showed that the OEE inhibits the iNOS in a dose-dependent manner, giving a 24% enzyme inhibition at the lowest dose (0.1 mg/mL) and 86% enzyme inhibition at the highest dose (10.0 mg/mL). From the comparison of the IC_50_ values, an isoform selectivity emerged, since OEE showed to be approximately three folds more potent against the iNOS with respect to the eNOS, with IC_50_ values of 0.485 and 1.460 mg/mL, respectively. On the contrary, the pure HSE was not able to interact with both the iNOS and eNOS, as their enzymatic activity was not affected.

Very interestingly, the HM showed a dose-dependent inhibition of iNOS, ranging from 11% at 0.1 mg/mL to 67% at 10 mg/mL, with an IC_50_ = 0.743 mg/mL. This quite potent anti-iNOS activity suggests that the HM could be endowed with a promising anti-inflammatory activity. Regarding its selectivity with respect to eNOS, it was complete only for 0.1 and 0.5 mg/mL, whereas, at higher concentrations, the HM showed non-selective effects. In particular, the eNOS IC_50_ was 2.051 mg/mL, revealing that the HM was more than 2.5 folds more potent against the iNOS with respect to the constitutive isoform (eNOS/iNOS selectivity ratio = 2760 folds). These results suggest that the HM could show a beneficial, selective iNOS inhibition up to 1.0 mg/mL, without potential cardiovascular side effects due to eNOS inhibition.

The relationship between inflammation and cancer formation is supported by a large amount of evidence [51]. IBD is a crucial risk factor for CRC, and inflammation is probably a trigger component of other forms of sporadic, as well as heritable, colon cancer. Phytocomplexes exert anticancer effects by many different mechanisms, including the promotion of apoptosis [52]. Moreover, drugs developed from natural sources are generally provided with better bioactivities and lower toxicity, and some of them have been administered along with conventional therapies to boost cancer cells’ vulnerability. Among these, for example, oleuropein exerts an anticancer activity, alone or in combination with conventional anticancer treatments, through different mechanisms in different cancer cell lines [53,54]. Moreover, OEE-derived compounds have a favorable effect against CRC, by also modulating gut microbiota, in reducing crypt dysplasia, and possess protective effects in colitis-associated CRC in vivo [55,56]. On the other hand, the anthocyanins also found in HSE possess antitumoral and anti-inflammatory properties [57]. In this view, the potential of the HM towards CRC was assessed. Interestingly, the HM caused apoptosis of Caco-2 cells with a 24 h-IC_50_ of ~250 µg/mL, and this effect was irreversible, suggesting that the HM triggered the so-called “point of no return”, a limit where irreversible damage occurs. We also checked whether the observed overall effect of the HM is a result of an additive- (i.e., the combined effect is a pure summation effect), antagonistic- (i.e., the combined effect is lower than the summation) or synergic (i.e., combined effect is greater than the additive) activity between the *Olea europea* L. (OEE) and *Hibiscus sabdariffa* L. (HSE) components. A synergist effect was demonstrated, as the cytotoxicity caused by the HM was always significantly higher than that elicited by the single OEE and HSE, as already observed at the cardiovascular level [8]. 

The present results also suggest that the activity of the HM was due to all its components, and not only to oleuropein-like compounds. The components from HSE, in fact, might play a role in the pro-apoptotic effects of HM. Thus, it will be interesting to assess how and which HSE components (hibiscus acid?) promote synergic effects with those of the OE leaves extract.

Regarding the mechanisms according to which HM exerts antitumoral activity towards Caco-2 cells, we can hypothesize a pro-oxidant effect triggered by the contained polyphenols, a key point for the activation of the apoptotic process. Interestingly, oleuropein can be a pro-oxidant in vitro in the same range of concentrations used in the present study [58]. At the base of the observed effects, however, other mechanisms cannot be excluded. For example, oleuropein possesses anticancer activity because of the down-regulation of the expression of BCL-2 and COX-2 proteins, and the inhibition of the multiple stages in colon carcinogenesis [59]. On the other hand, HSE also possesses antitumoral activity, arising via both the activation of p38 MAP kinase and a ROS-mediated mitochondrial dysfunction pathway [60]. Moreover, an in vivo antileukemic activity of HSE has also been reported [61]. 

The healthy-tissue-sparing activity of the HM was also assessed, and the results on the rat ileum and proximal colon highlighted a safe profile of the HM up to 1.0 mg/mL, suggesting a selective activity against cancer cells at the concentrations needed to induce apoptosis. This sparing activity was already reported for other polyphenols [62], including those of *Olea europea* L. leaves extracts [63]. Finally, the fact that the HM can selectively target iNOS in the concentration range that causes Caco-2 cell apoptosis is an added value that strengthened the use of this mix to prevent/treat the inflammatory-based disease of the intestine.

## 5. Conclusions

In the nutraceutical field, the assessment of the role played by the different target networks connected to pathologies becomes very important in order to ensure the effectiveness of phytocomplexes, where the presence of pools of organic molecules can act on different nodes of the network itself.

By examining in depth the off-target activity of the HM, we have shown that it can control the ileum and colon contractility without blocking the progression of the food bolus, and that it possesses a strong pro-apoptotic activity on CRC cells, along with selective iNOS inhibition. These results suggest that the HM can be envisaged as a formulation that is potentially useful in the modulation of some targets linked to inflammatory-based intestinal disease. To achieve this goal, however, it is also important to assess which is the best ratio between OEE and HSE that gives rise to the most powerful effects, and to set up a formulation that selectively targets the intestine. Moreover, our results suggest that, in some diseases, such as those related to motility disorders, an appropriate nutritional approach can be accompanied by a correct use of nutraceuticals that could help not only in ameliorating the symptoms, as documented for the use of phytochemicals in IBD [64], but also in preventing more severe conditions, such as CRC.

## Figures and Tables

**Figure 1 nutrients-14-00463-f001:**
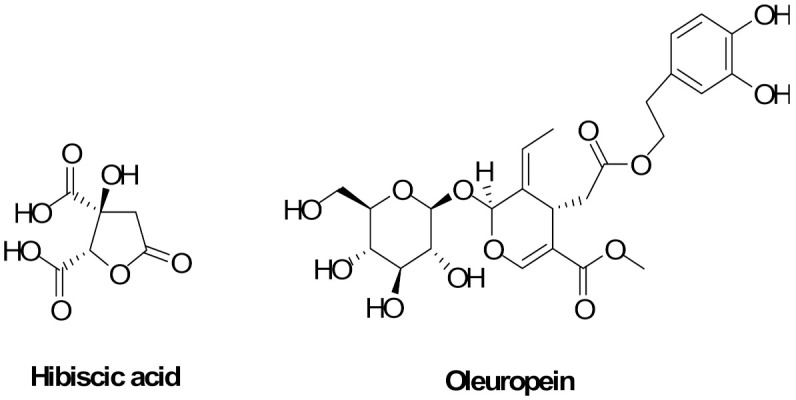
Chemical structures of the most representative chemicals found in HM (Herbal Mix): hibiscus acid and oleuropein.

**Figure 2 nutrients-14-00463-f002:**
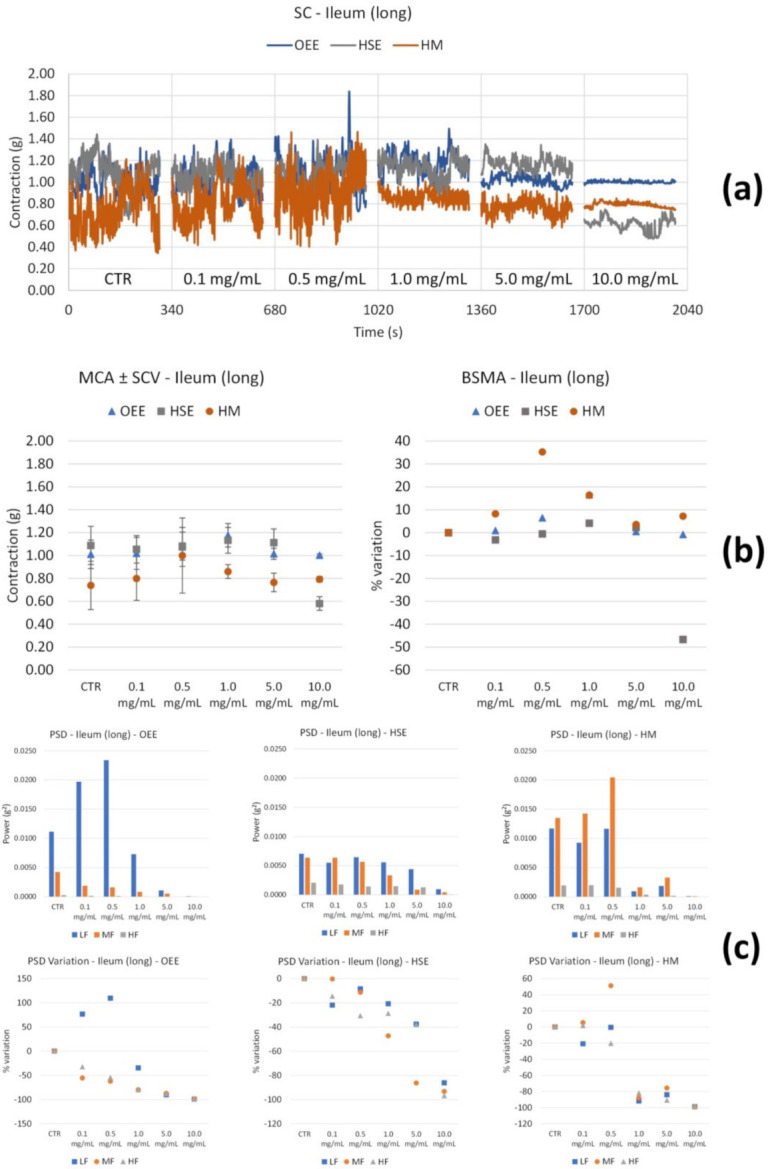
Spontaneous ileum longitudinal basal contractility recordings of the concentration-response curve of OEE (*Olea europea* L. leaf), HSE (*Hibiscus sabdariffa* L.) and HM (Herbal Mix). (**a**) Spontaneous contraction (SC) signals for each concentration; (**b**) mean contraction amplitude (MCA) and spontaneous contraction variability (SCV), represented as error bars in the MCA plot and contraction percentage variation with respect to the control (BSMA) for each considered condition; (**c**) absolute powers (PSD) of the different bands of interest (LF: [0.0,0.2[ Hz; MF: [0.2,0.6[ Hz; HF: [0.6,1.0] Hz) and PSD% variations with respect to the control phase.

**Figure 3 nutrients-14-00463-f003:**
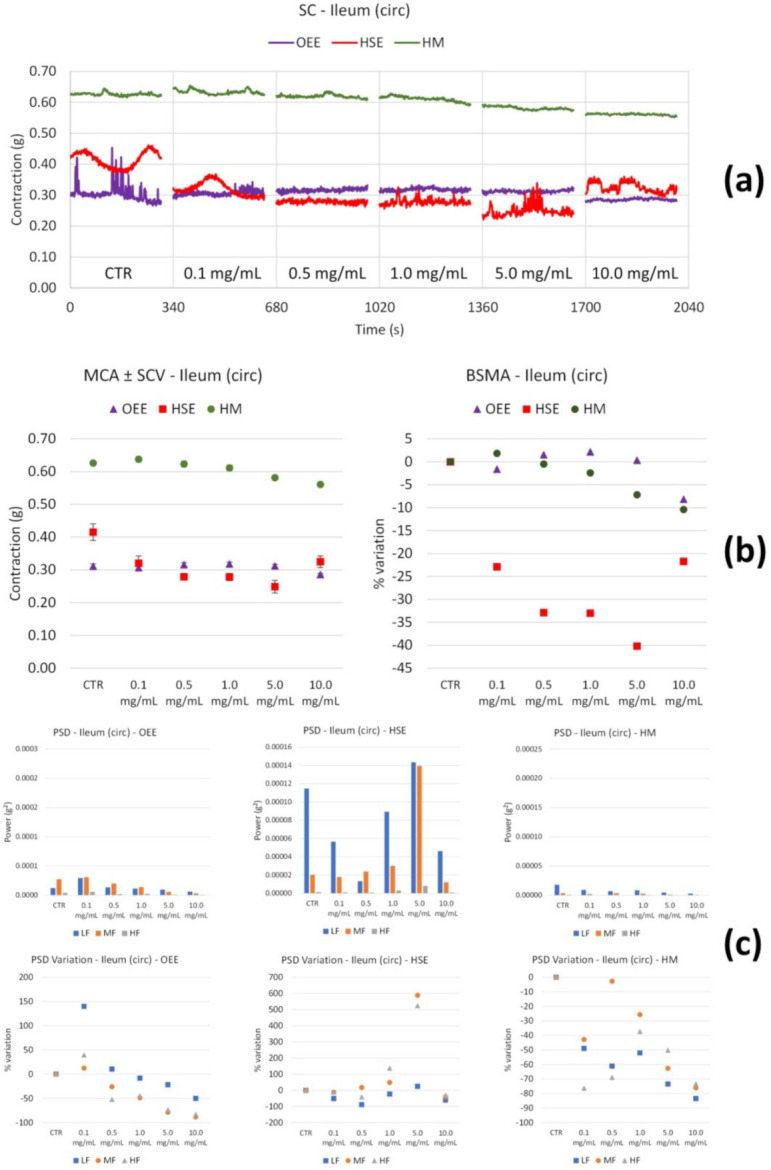
Spontaneous ileum circular basal contractility recordings of the concentration-response curve of OEE (*Olea europea* L. leaf), HSE (*Hibiscus sabdariffa* L.) and HM (Herbal Mix). (**a**) Spontaneous contraction (SC) signals for each concentration; (**b**) mean contraction amplitude (MCA) and spontaneous contraction variability (SCV), represented as error bars in the MCA plot and contraction percentage variation with respect to the control (BSMA) for each considered condition; (**c**) absolute powers (PSD) of the different bands of interest (LF: [0.0,0.2[ Hz; MF: [0.2,0.6[ Hz; HF: [0.6,1.0] Hz) and PSD % variations with respect to the control phase.

**Figure 4 nutrients-14-00463-f004:**
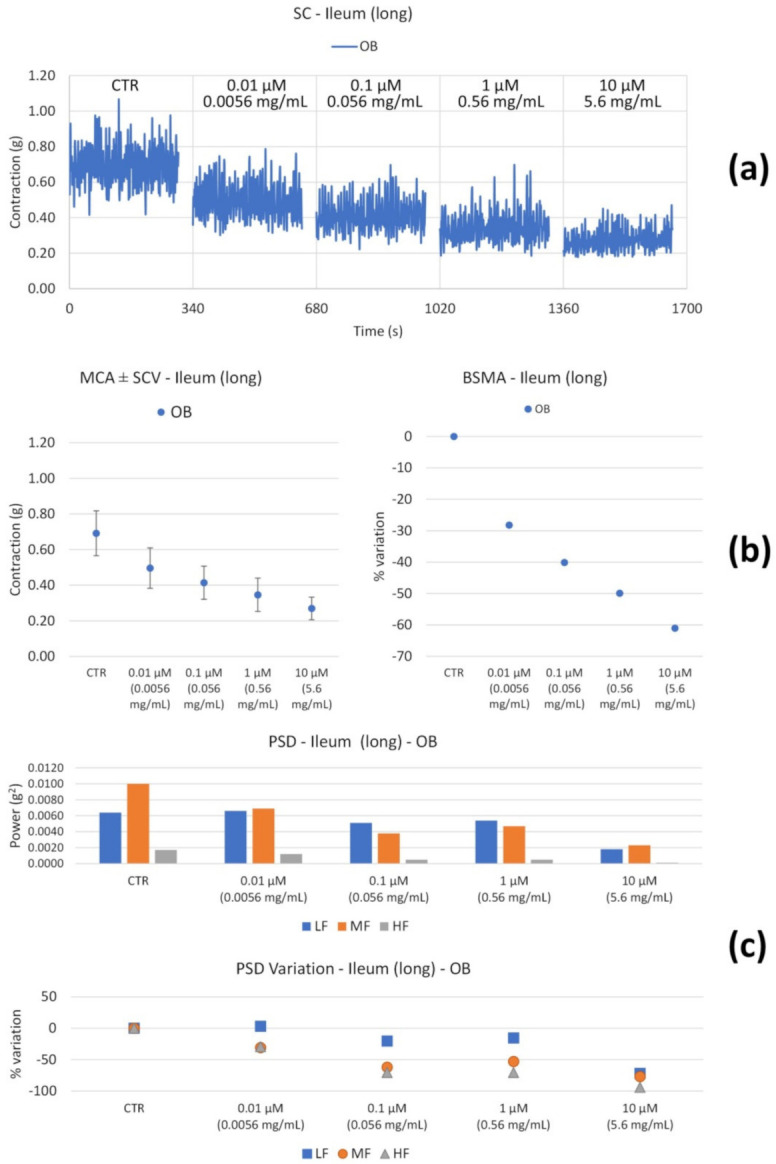
Spontaneous ileum longitudinal basal contractility recordings of the concentration-response curve of OB (Otylonium Bromide). (**a**) Spontaneous contraction (SC) signals for each concentration; (**b**) mean contraction amplitude (MCA) and spontaneous contraction variability (SCV), represented as error bars in the MCA plot and contraction percentage variation with respect to the control (BSMA) for each considered condition; (**c**) absolute powers (PSD) of the different bands of interest (LF: [0.0,0.2[ Hz; MF: [0.2,0.6[ Hz; HF: [0.6,1.0] Hz) and PSD% variations with respect to the control phase.

**Figure 5 nutrients-14-00463-f005:**
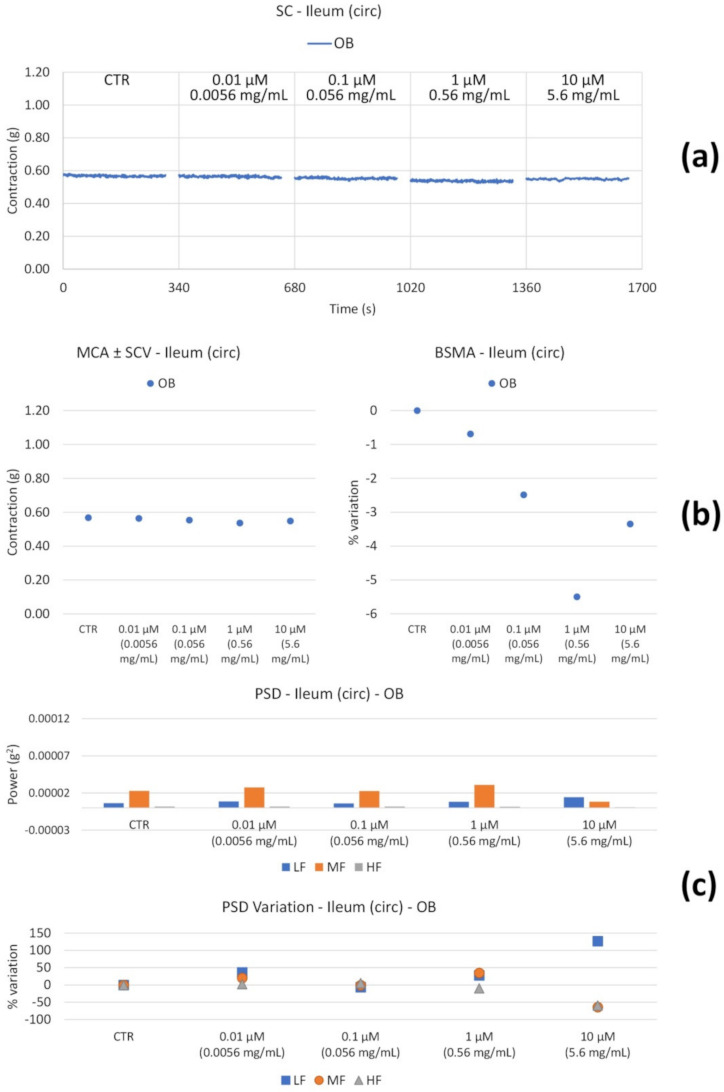
Spontaneous ileum circular basal contractility recordings of the concentration-response curve of OB (Otylonium Bromide). (**a**) Spontaneous contraction (SC) signals for each concentration; (**b**) mean contraction amplitude (MCA) and spontaneous contraction variability (SCV), represented as error bars in the MCA plot and contraction percentage variation with respect to the control (BSMA) for each considered condition; (**c**) absolute powers (PSD) of the different bands of interest (LF: [0.0,0.2[ Hz; MF: [0.2,0.6[ Hz; HF: [0.6,1.0] Hz) and PSD% variations with respect to the control phase.

**Figure 6 nutrients-14-00463-f006:**
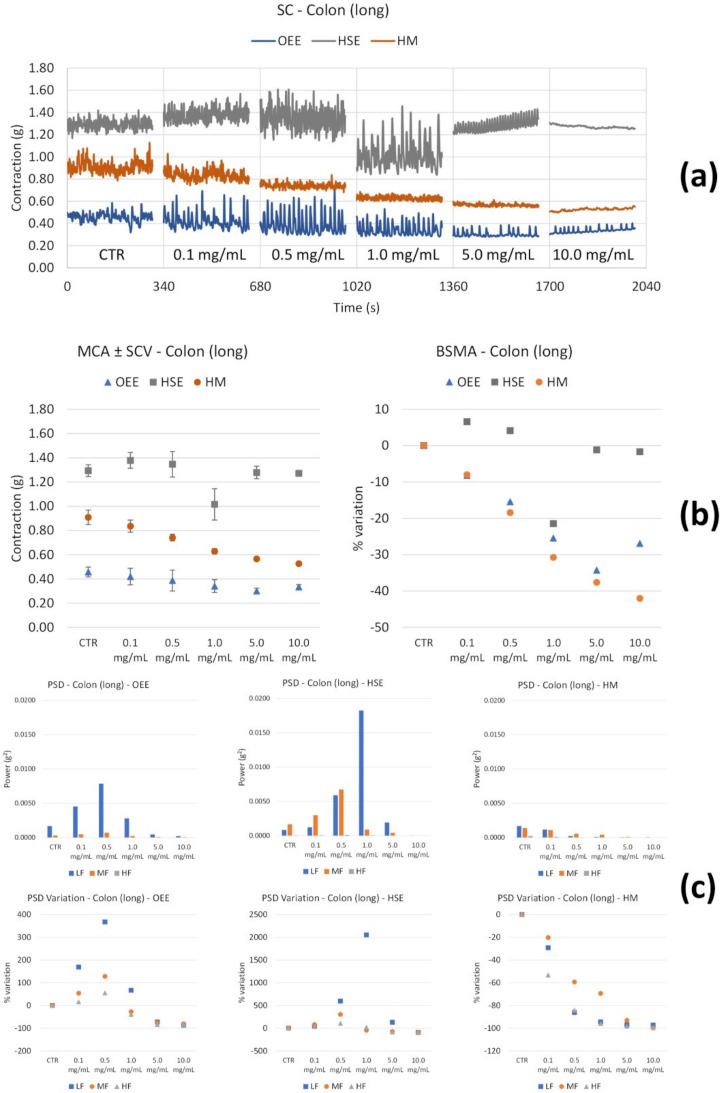
Spontaneous colon longitudinal basal contractility recordings of the concentration-response curve of OEE (*Olea europea* L. leaf), HSE (*Hibiscus sabdariffa* L.) and HM (Herbal Mix). (**a**) Spontaneous contraction (SC) signals for each concentration; (**b**) mean contraction amplitude (MCA) and spontaneous contraction variability (SCV), represented as error bars in the MCA plot and contraction percentage variation with respect to the control (BSMA) for each considered condition; (**c**) absolute powers (PSD) of the different bands of interest (LF: [0.0,0.2[ Hz; MF: [0.2,0.6[ Hz; HF: [0.6,1.0] Hz) and PSD% variations with respect to the control phase.

**Figure 7 nutrients-14-00463-f007:**
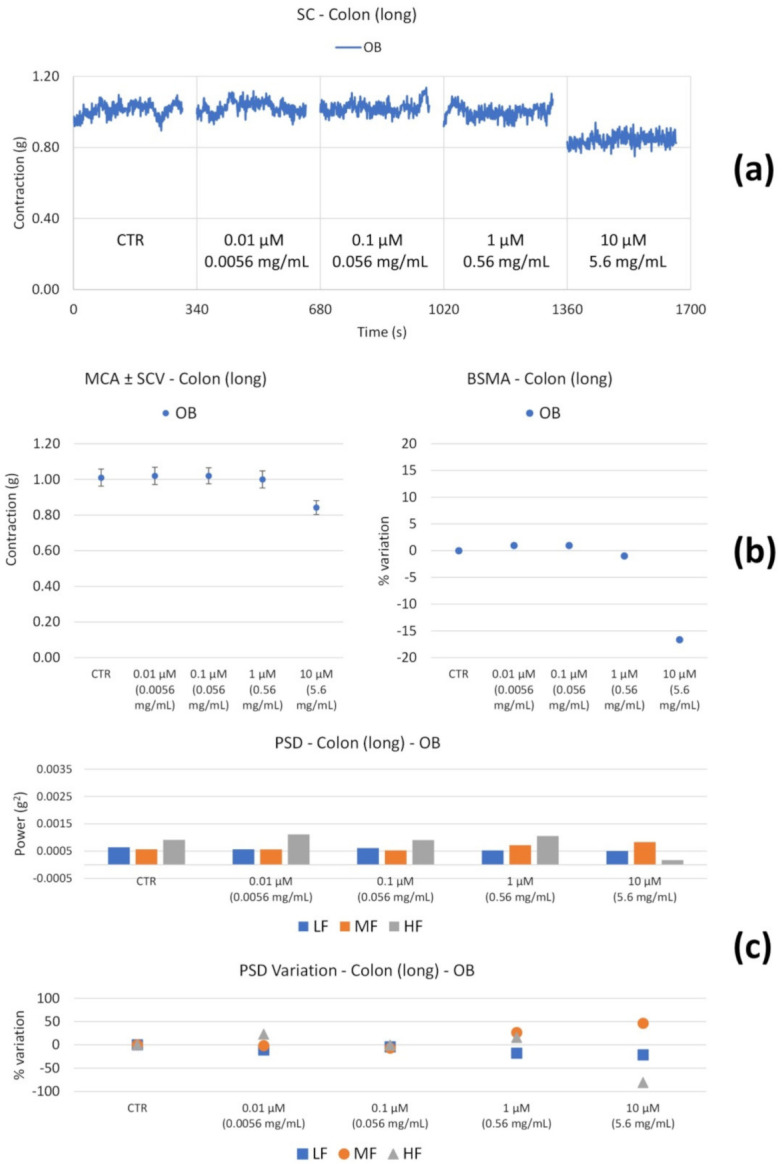
Spontaneous colon longitudinal basal contractility recordings of the concentration-response curve of OB (Otylonium Bromide). (**a**) Spontaneous contraction (SC) signals for each concentration; (**b**) mean contraction amplitude (MCA) and spontaneous contraction variability (SCV), represented as error bars in the MCA plot and contraction percentage variation with respect to the control (BSMA) for each considered condition; (**c**) absolute powers (PSD) of the different bands of interest (LF: [0.0,0.2[ Hz; MF: [0.2,0.6[ Hz; HF: [0.6,1.0] Hz) and PSD% variations with respect to the control phase.

**Figure 8 nutrients-14-00463-f008:**
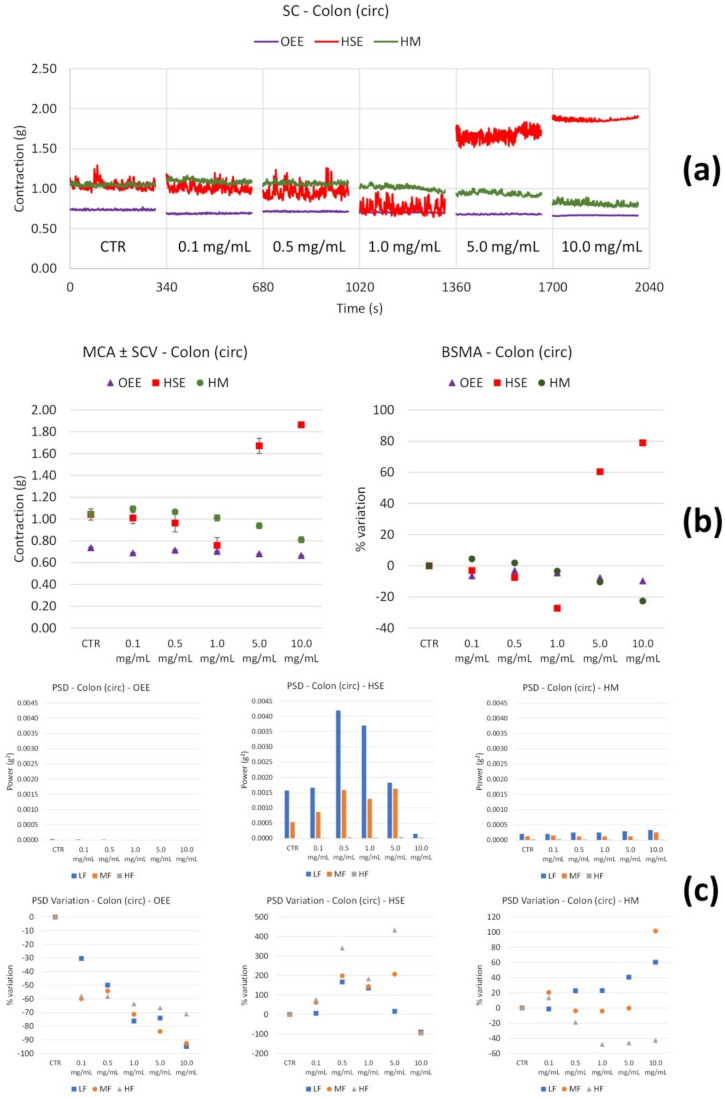
Spontaneous colon circular basal contractility recordings of the concentration-response curve of OEE (*Olea europea* L. leaf), HSE (*Hibiscus sabdariffa* L.) and HM (Herbal Mix). (**a**) Spontaneous contraction (SC) signals for each concentration; (**b**) mean contraction amplitude (MCA) and spontaneous contraction variability (SCV), represented as error bars in the MCA plot and contractions percentage variation with respect to the control (BSMA) for each considered condition; (**c**) absolute powers (PSD) of the different bands of interest (LF: [0.0,0.2[ Hz; MF: [0.2,0.6[ Hz; HF: [0.6,1.0] Hz) and PSD % variations with respect to the control phase.

**Figure 9 nutrients-14-00463-f009:**
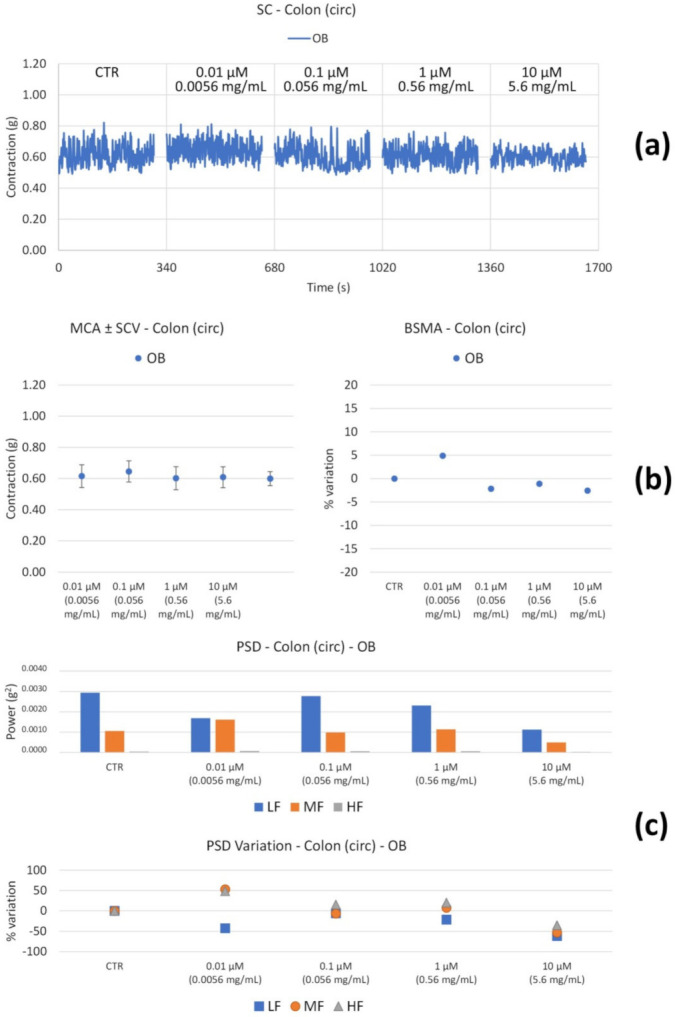
Spontaneous colon circular basal contractility recordings of the concentration-response curve of OB (Otylonium Bromide). (**a**) Spontaneous contraction (SC) signals for each concentration; (**b**) mean contraction amplitude (MCA) and spontaneous contraction variability (SCV), represented as error bars in the MCA plot and contraction percentage variation with respect to the control (BSMA) for each considered condition; (**c**) absolute powers (PSD) of the different bands of interest (LF: [0.0,0.2[ Hz; MF: [0.2,0.6[ Hz; HF: [0.6,1.0] Hz) and PSD % variations with respect to the control phase.

**Figure 10 nutrients-14-00463-f010:**
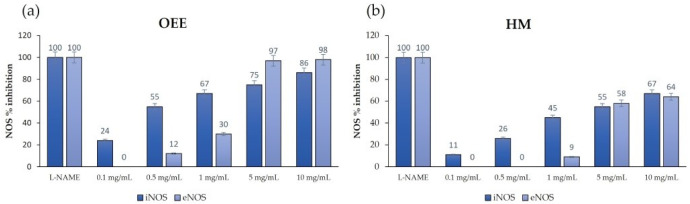
iNOS and eNOS inhibition by OEE (**a**) and HM (**b**). Results represent the mean ± SD of three independent experiments. Control NOS reactions were performed in the absence of inhibitor (0% inhibition), and L-NAME 1 mM was used as the positive control (100% inhibition).

**Figure 11 nutrients-14-00463-f011:**
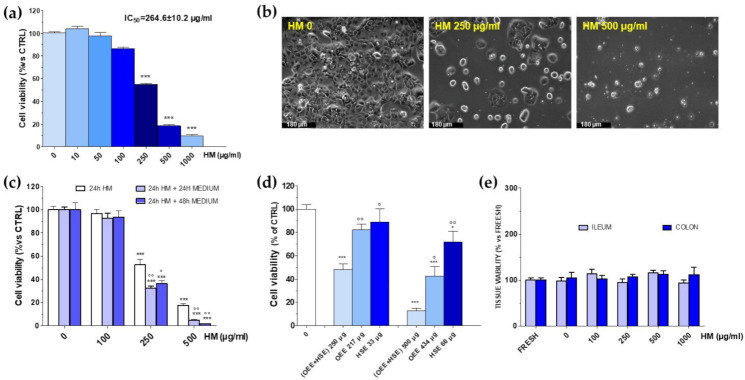
Human colorectal adenocarcinoma Caco-2 cells’ viability after treatment for 24 h with increasing concentration of herbal mix (HM, 0–1000 µg/mL) (**a**). Contrast-phase microscopy morphological analysis performed in untreated (HM 0 µg/mL) and HM-treated (250–500 µg/mL, 24 h) Caco-2 cells (scale bar 180 µm). Each photograph was representative of three independent observations (**b**). Reversibility of cytotoxic effect caused by HM (**c**). Caco-2 cells’ viability after treatment with 250 µg/mL or 500 µg/mL of HM (OEE + HSE) or to the corresponding amount found in 250 µg/mL or 500 µg/mL of only the *Olea europea* L. leaves extract (OEE, 217 and 433 µg/mL, respectively) or *Hibiscus sabdariffa* L. calyces (HSE, 33 and 67 µg/mL, respectively) (**d**). Rat ileum and proximal colon rings’ viabilities after treatment with HM for 24 h. FRESH: tissue immediately after being explanted. HM 0 µg/mL: tissue treated with PSS for 24 h. In all panels, values are mean ± SEMs of four or five independent experiments run in quadruplicate; controls (HM 0 µg/mL) represent untreated cells (**e**). Statistical significance was assessed by ANOVA followed by Dunnett or Bonferroni post-test. (**a**): *** *p* < 0.001 vs. controls (HM 0 µg/mL); (**c**): *** *p* < 0.001 vs. controls (HM 0 µg/mL); ° *p* < 0.05, °° *p* < 0.01 vs. 24 h HM same concentration; (**d**) *** *p* < 0.001, * *p*<0.05 vs. controls (HM 0 µg/mL), ° *p* < 0.05, °° *p* < 0.01 vs. (OEE + HSE) same bar group.

**Figure 12 nutrients-14-00463-f012:**
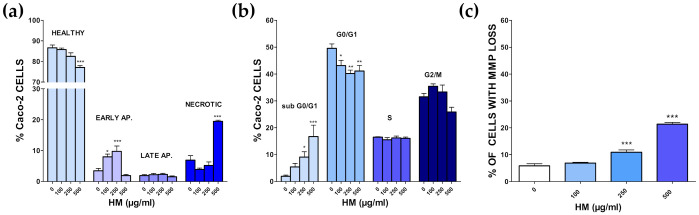
HM-mediated effects on Caco-2 cell cycle. (**a**). Annexin V and propidium iodide double staining assay (**b**). Cell cycle analysis. (**c**) Percentage of cells with loss in mitochondria membrane potential (Rhodamine123 staining). Values are mean ± SEMs of at least four independent experiments in which three points/concentrations were run; controls (HM 0 µg/mL) represent untreated cells. * *p* < 0.05, ** *p* < 0.01, *** *p* < 0.001 vs. controls (ANOVA followed by Dunnett post-test).

**Table 1 nutrients-14-00463-t001:** Quantitative analysis of phenolic compounds in HSE and OEE determined by HPLC-MS/MS analysis modified from [8].

Compound	HSE ^a^	OEE ^a^
Hibiscus acid	139.20 ± 0.47	
Secoiridoids ^b^		309.83 ± 0.65
Hydroxycinnamic acids		4.67 ± 0.44
Flavonols		0.69 ± 0.06
Flavones		10.77 ± 0.93

^a^ Data are expressed as mg/g. ^b^ Among these, oleuropein and its isomers represent 85.91%. HSE: *Hibiscus sabdariffa* L., OEE: *Olea europea* L. leaf.

**Table 2 nutrients-14-00463-t002:** Spasmolytic activity of HM and its components on K^+^-depolarized guinea pig ileum and colon longitudinal smooth muscle.

	Ileum			Colon		
	% Intrinsic Activity ^a^	IC_50_ ^b^ (mg/mL)	95% C.L.	% Intrinsic Activity ^a^	IC_50_ ^b^ (mg/mL)	95% C.L.
**OEE**	90.0 ± 1.7 (10.0 mg/mL)	0.77	0.68−0.89	75.0 ± 2.4 (0.4 mg/mL)	0.13	0.08−0.20
**HSE**	21.0 ± 0.7 (5.0 mg/mL)			20.0 ± 1.3 (10.0 mg/mL)		
**HM**	90.0 ± 1.2 (10.0 mg/mL)	1.39	1.13−1.54	85.0 ± 1.3 (1.0 mg/mL)	0.28	0.22−0.36
**OB ** ^ **#** ^	90 ± 3 (0.0056 mg/mL, 10 µM)	0.0048 (8.52 µM)	0.0040−0.0057 (7.14−10.11 µM)	90 ± 2.3 (0.02 mg/mL, 50 µM)	0.0019 (3.43 µM)	0.0015−0.0025 (2.65−4.44 µM)

Data are reported as mean ± SEM, while C.L. represent the confidence limits. ^a^ Percentage inhibition of calcium-induced contraction on K^+^-depolarized (80 mM) guinea pig ileum and colon longitudinal smooth muscle. The concentration that gave the maximum effect is reported in parenthesis. ^b^ IC_50_ (i.e., the concentration that inhibited 50% of the maximum contraction induced by K+ 80 mM) was calculated according to [27] (*n* = 6–7). When the maximum effect was <50%, the IC_50_ values were not calculated. **^#^** The results of OB (Otylonium Bromide) were expressed in both mg/mL and μM to better compare the results. Data on ileum are from [9]. OEE: *Olea europea* L. leaf, HSE: *Hibiscus sabdariffa* L., HM: Herbal Mix.

**Table 3 nutrients-14-00463-t003:** Outlook of the effects of OEE, HSE and HM on guinea-pig ileum smooth muscle spontaneous contractility.

Ileum		Tone	Variability
**OEE**	Long	↑ up to 5.0 mg/mL	↓↓↓ LF from 1.0 mg/mL
	Circ	↔	↑ LF and MF (only at 0.1 mg/mL), then ↓
**HSE**	Long	↓↓↓ only at 10.0 mg/mL	↓ all the FB (only at 10.0 mg/mL)
	Circ	↓ with concentration, maximum effect at 5.0 mg/mL	↑↑↑ LF and MF (at 5.0 mg/mL)
**HM**	Long	↑ up to 0.5 mg/mL, then ↓	↔ up to 0.5 mg/mL, then ↓ at all the FB
	Circ	↔ up to 1.0 mg/mL, then ↓	↓ with concentration of all FB
**OB**	Long	↓ with concentration	Global reduction; MF and HF progressive ↓; ↓↓↓ LF only at high concentrations.
	Circ	↔	↔

Legend: ↑, increase; ↑↑↑, significant increase; ↓, decrease; ↓↓↓, significant decrease: ↔, constant; FB, Frequency Band. OEE: *Olea europea* L. leaf, HSE: *Hibiscus sabdariffa* L., HM: Herbal Mix, OB: Otylonium Bromide, LF: Low Frequencies, MF: Medium Frequencies, FB: Frequency Band, HF: High Frequencies.

**Table 4 nutrients-14-00463-t004:** Outlook of the effects of OEE, HSE and HM on guinea-pig colon smooth muscle spontaneous contractility.

Colon		Tone	Variability
**OEE**	Long	↓ maximal effect at 5.0 mg/mL	↑ LF and MF bands (up to 0.5 mg/mL), then ↓
	Circ	↓ (very weak) with concentration	↓ of all the FB with concentration
**HSE**	Long	↓ only at 1.0 mg/mL	↑ LF (up to 1.0 mg/mL); ↑MF (up to 0.5 mg/mL); ↓ of ll the FB for higher frequencies.
	Circ	↓ (up to 1.0 mg/mL), then ↑↑↑	↑↑↑ of LF and MF (0.5 and 1.0 mg/mL): (at 10.0 mg/mL)
**HM**	Long	↓ with concentration	↓ with concentrations of all FB
	Circ	↓ (weak) with concentration	↑ LF with concentration; MF ↑ (at 10.0 mg/mL)
**OB**	Long	↓ (only at 0.0056 mg/mL)	↓HF (at 0.0056 mg/mL)
	Circ	↔	↓ (weak) LF (at 0.0056 mg/mL)

Legend: ↑, increase; ↑↑↑, significant increase; ↓, decrease; ↔, constant; FB: Frequency Band.

**Table 5 nutrients-14-00463-t005:** Cardiovascular activity of HM and its components.

	Left Atrium			Right Atrium						Aorta		
	Negative Inotropy			Negative Inotropy			Negative Chronotropy			Vasorelaxant		
	Intrinsic Activity ^a^	EC_50_ ^b^	95% C.L.	Intrinsic Activity ^c^	EC_50_ ^b^	95% C.L.	Intrinsic Activity ^d^	EC_50_ ^b^	95% C.L.	Intrinsic Activity ^e^	EC_50_ ^b^	95% C.L.
**OEE**	68.0 ± 2.4 (*1*)	0.14	0.10–0.18	**81.0 ± 1.5 (*1*)**	**0.35**	**0.28–0.35**	37.0 ± 2.4 (*10*)			90.0 ± 1.1 (*10*)	5.15	4.68–5.59
**HSE**	76.0 ± 0.9 (*1*)	0.27	0.21–0.35	**71.0 ± 2.4 (*1*)**	**0.26**	**0.19–0.34**	46.0 ± 0.7 (*1*)			93.0± 1.4 (*10*)	6.63	6.34–6.92
**HM**	60.0 ± 1.4 (*1.0*)	0.16	0.12–0.20				84.0 ± 2.0 (*10*)	1.21	1.10–1.33	70.0 ± 3.6 (*10*)	5.89	5.56–6.25
**OB ^#^**	**92.0 ± 1.6**			**69.0 ± 1.6**			**1.0 ± 0.1**			**5.0 ± 0.2**		
	**(0.00056)**	0.000021	0.000019–0.000022	**(0.0056)**	**0.000027**	**0.000021–0.000031**	**(0.056)**			**(0.056)**		
	**(*1*)**	**0.037**	**0.034–0.039**	**(*10*)**	**0.049**	**0.037–0.055**	**(*100*)**			**(*100*)**		

Data are reported as mean ± SEM, while C.L. represent the confidence limits. ^a c^ Decrease in developed tension on isolated guinea-pig left atrium (a) and on guinea-pig spontaneously beating isolated right atrium (c) at the indicated concentration in parenthesis (mg/mL) that gave the maximum effect, expressed as percentage changes from the control (*n* = 5–6). The left atria were driven at 1 Hz. ^b^ Calculated from concentration–response curves according to [27] (*n* = 6–7). When the maximum effect was <50%, the EC_50_ inotropic, EC_50_ chronotropic and IC_50_ vasorelaxant values were not calculated. ^d^ Decrease in atrial rate on guinea-pig spontaneously beating isolated right atrium at the indicated concentration in parenthesis (mg/mL), expressed as percentage changes from the control (*n* = 7–8). The indicated concentration gave the maximum effect. Pretreatment heart rate ranged from 165 to 190 beats/min. ^e^ Percentage inhibition of calcium-induced contraction on K^+^-depolarized (80 mM) guinea pig aortic strip smooth muscle at the indicated concentration. The concentration that gave the maximum effect is in parenthesis and expressed as mg/mL. **^#^** The results of OB were expressed in both mg/mL and μM to better compare the results. Bold: unpublished data.

**Table 6 nutrients-14-00463-t006:** iNOS and eNOS IC_50_ values ± SD calculated from concentration–response curves of three independent experiments [15].

	iNOS	eNOS	eNOS/iNOS Selectivity
	IC_50_ (mg/mL)		
**OEE**	0.485 ± 0.024	1.460 ± 0.071	3.140
**HM**	0.743 ± 0.029	2.051 ± 0.094	2.760

## Data Availability

Not applicable.

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
