# Peer review of "Olea europea L. Leaves and Hibiscus sabdariffa L. Petals Extracts: Herbal Mix from Cardiovascular Network Target to Gut Motility Dysfunction Application"

_nutrients, 2022, doi:10.3390/nu14030463_

Round 1

Reviewer 1 Report

The Authors submitted a paper regarding the biological effects (cardiovascular, gut motility) of a herbal mix. The work is interesting and well written. However, I suggest to the Authors add some information regarding the cytotoxic effect of the extracts on non-cancerous cells. Besides, a more detailed chemical analysis should be reported as supplementary material.

Line 61: lives - leaves

Line 91: Please, add plant extract code.

Line 199: Did the Authors perform citotoxicity assay on non cancerous cells? If so, the results should be reported.

Line 252: I suggest to report a more detailed chemical analysis as supplementary material.

Page 9: Image resolution should be improved.

Author Response

                                                                                                                      Bologna, 09 January-2022

Revision of Manuscript: ID nutrients-1524439

Olea europea L. Leaves and Hibiscus sabdariffa L. petals Extracts: herbal mix from cardiovascular network target to gut motility dysfunction application”

by: Laura Beatrice Mattioli, Maria Frosini, Rosa Amoroso, Cristina Maccallini, Elda Chiano, Rita Aldini, Francesco Urso, Ivan Corazza, Matteo Micucci, and Roberta Budriesi

Dear Editor,

We have carefully read the comments of the Reviewers and we are grateful for the useful criticisms that allowed us to improve the quality of this study. The manuscript has been extensively revised, addressing all the points raised by the referees. Below we report a point-to-point reply to each specific criticism raised. We hope that in the present form it is suitable for publication in the Special Issue "The Role of Food-Derived Bioactive Compounds in Modulating Inflammasome".

Thank you again for your attention and we look forward to hearing from you soon.

                                                                                                         Sincerely

                                                                                               Roberta Budriesi Prof.

Reviewer 1:

Comments and Suggestions for Authors

The Authors submitted a paper regarding the biological effects (cardiovascular, gut motility) of a herbal mix. The work is interesting and well written. However, I suggest to the Authors add some information regarding the cytotoxic effect of the extracts on non-cancerous cells.

Besides, a more detailed chemical analysis should be reported as supplementary material.

We agree with the referee on the opportunity to include more information about the plants used, the extraction methods and the analytical techniques used for the characterization. This research is the continuation of previous works (please see PubMed Budriesi Roberta) aimed at exploring various target networks connected to various pathologies as a result of the biological actions highlighted where the same extracts and the same herbal mix were used to acquire more information and target (as for drugs) potential applications. For these reasons it did not seem appropriate to repeat similar information here as well, with the risk of running into problems relating to plagiarism. However, we have included a further and very recent [MDPI 2021] open access bibliography where there are some details and all the indications to find the information required to facilitate any further study by the reader.

Line 61: lives – leaves

The correction has been done

Line 91: Please, add plant extract code.

See answer above regarding the details of the materials and methods used.

Line 199: Did the Authors perform citotoxicity assay on non cancerous cells? If so, the results should be reported.

We thank the Referee for this suggestion. Indeed, such experiments have not been performed as we preferred to test the effects of HM on a more complex and physiological context such as the whole healthy tissue. Orally administered drugs are exposed to a changing environment, and even they are planned to target the colon, they will be in direct contact with both the upper part of the GI tract and colon itself. It is therefore mandatory that healthy tissue is not affected by the treatment. For this reason, we tested the effects of 24h treatment with HM on rat colon ring viability. Ileum rings were tested as well, as small intestine tissue constitutes the largest part of the gastro-intestinal tract. The results reported in the manuscript demonstrated that HM did not affect tissue viability of ileum and proximal colon rings, suggesting a safe profile of HM up to 1 mg/mL. This concentration is four-two time higher than those required to exert apoptosis of cancer cells (250-500 µg/mL), suggesting a safe profile of HM. This sparing activity was already reported for other polyphenols [62], including those of Olea europea L. leaves extracts [63].

Line 252: I suggest to report a more detailed chemical analysis as supplementary material.

See answer above regarding the details of the materials and methods used

Page 9: Image resolution should be improved.

All imagines were been checked and figure 9 has been changed.

Reviewer 2 Report

This manuscript presents various analyses of Hibiscus sabdariffa L. (HSE) extract, Olea Europea L. (OEE) extract, and their combination “Herbal Mix” (HM), as well as Otylonium bromide (OB) on guinea-pig ileum and colon contractility, human iNOS/eNOS activity, and colorectal adenocarcinoma via the analysis of Caco-2 cells.

General Comments:

The comparison being used to determine differences and changes is often unclear. Was the positive control (OB) always being used as the comparator, or in most instances where a comparator is not noted is a statistical association rather than effect being assessed?

How many samples were assessed for each analysis? Could there be variability depending on the sample size that may affect the findings?

The figures do not appear to be referred to in numerical (nor alphabetical, i.e., Figure 11c vs Figure 11a) order within the manuscript text. Suggest updating to ensure the appropriate order within the text.

Suggest revising for grammar and spelling, as well as writing all acronyms in full the first time they appear in the manuscript (ex. PSS, AV, PI, LTCC, etc.), as well as in tables and figures.

Specific Comments:

Figures 2-9: These figures would benefit from greater clarity in their presentation as there appears to be numerous subfigures within each figure. Suggest either visually summarizing the data more succinctly or adding letters to further indicate the specific figure that is being referred to within the text as currently it is unclear which graphical representation of the data the statements in the manuscript are referring to and these statements do not always appear to be clearly supported by the data presented. For example, such as how Figures 10-12 are presented with letters within each Figure for each graphical representation of the data. As well, ensure the description for each figure contains all relevant details so that the figure may be understandable as stand-alone content.

Figures 3, 5, 7, and 9: Since it is noted in the manuscript that OB is acting as a control, it is unclear why this data is presented in separate figures. As the positive control, would this not be the comparator for the experimental interventions (OEE, HSE, HM) and thus more applicable to be included in the figures with this data for comparison?

Line 390: It is unclear why OB is being utilized as a positive control in this scenario given it is noted in lines 386-387 that the effect of OB on cardiovascular parameters was assessed, suggesting the effect is currently unknown. Whereas a positive control is a treatment that is known to produce the expected effect to validate the experimental procedure. Since it seems as though OB is being treated as an intervention, was a control used for this assessment?

Lines 400-411: This appears to be a description of methods; however, it is currently noted in the results section. Suggest revising the text to include this description in the methods section. (An additional minor note, L-citrulline is written as L-Cit later in line 408, yet this abbreviation was not previously indicated and does not seem necessary in this context.)

Line 420: Could this line please be clarified, specifically the determination and representation of the 12% and 35%?

Lines 477-494: This section reads like content more applicable to a discussion section, suggest modifying accordingly.

Lines 518-521: Similar to a note above, it appears as though this results section is currently presenting a description of methodologies utilized as well as conjecture more relevant to a discussion section. Hence, this content, may be more beneficially presented in the respective methods and discussion sections.

Lines 554-555: Could it please be clarified as to what “HM’s action trend is similar” to?

Lines 594-597: This statement does not appear to be verified by the present findings, and there does not appear to be a reference to support this as it relates to HM, thus perhaps this should be presented as more of a possibility (using could or may phrasing) rather than a statement.

Lines 683-686: This paragraph appears to be a duplicate of the paragraph above. Please remove.

Thank-you for your time and consideration of these suggestions.

Author Response

                                                                                                                      Bologna, 09 January-2022

Revision of Manuscript: ID nutrients-1524439

Olea europea L. Leaves and Hibiscus sabdariffa L. petals Extracts: herbal mix from cardiovascular network target to gut motility dysfunction application”

by: Laura Beatrice Mattioli, Maria Frosini, Rosa Amoroso, Cristina Maccallini, Elda Chiano, Rita Aldini, Francesco Urso, Ivan Corazza, Matteo Micucci, and Roberta Budriesi

Dear Editor,

We have carefully read the comments of the Reviewers and we are grateful for the useful criticisms that allowed us to improve the quality of this study. The manuscript has been extensively revised, addressing all the points raised by the referees. Below we report a point-to-point reply to each specific criticism raised. We hope that in the present form it is suitable for publication in the Special Issue "The Role of Food-Derived Bioactive Compounds in Modulating Inflammasome".

Thank you again for your attention and we look forward to hearing from you soon.

                                                                                                         Sincerely

                                                                                               Roberta Budriesi Prof.

Reviewer 2:

Comments and Suggestions for Authors

This manuscript presents various analyses of Hibiscus sabdariffa L. (HSE) extract, Olea Europea L. (OEE) extract, and their combination “Herbal Mix” (HM), as well as Otylonium bromide (OB) on guinea-pig ileum and colon contractility, human iNOS/eNOS activity, and colorectal adenocarcinoma via the analysis of Caco-2 cells.

General Comments:

The comparison being used to determine differences and changes is often unclear. Was the positive control (OB) always being used as the comparator, or in most instances where a comparator is not noted is a statistical association rather than effect being assessed?

How many samples were assessed for each analysis? Could there be variability depending on the sample size that may affect the findings?

The figures do not appear to be referred to in numerical (nor alphabetical, i.e., Figure 11c vs Figure 11a) order within the manuscript text. Suggest updating to ensure the appropriate order within the text.

All figures have been redone to increase the resolution and add more details for easier understanding.

Suggest revising for grammar and spelling, as well as writing all acronyms in full the first time they appear in the manuscript (ex. PSS, AV, PI, LTCC, etc.), as well as in tables and figures.

The referee's suggestion has been accepted and the related changes to the text were made.

Specific Comments:

Figures 2-9: These figures would benefit from greater clarity in their presentation as there appears to be numerous subfigures within each figure. Suggest either visually summarizing the data more succinctly or adding letters to further indicate the specific figure that is being referred to within the text as currently it is unclear which graphical representation of the data the statements in the manuscript are referring to and these statements do not always appear to be clearly supported by the data presented. For example, such as how Figures 10-12 are presented with letters within each Figure for each graphical representation of the data. As well, ensure the description for each figure contains all relevant details so that the figure may be understandable as stand-alone content.

Figures 3, 5, 7, and 9: Since it is noted in the manuscript that OB is acting as a control, it is unclear why this data is presented in separate figures. As the positive control, would this not be the comparator for the experimental interventions (OEE, HSE, HM) and thus more applicable to be included in the figures with this data for comparison?

We agree with the referee of the opportunity to have reference samples and compounds in the same figure. Moreover, this is always done if you compare samples and reference compounds by constructing the cumulative concentration-response curves using the same independent variable (e.g. molarity). On the other hand, we always do it as documented by literature (please see Budriesi Roberta on Pubmed). In this case, however, we have reference samples and compounds that are characterized by constructing cumulative concentration-response curves with different independent variables: mg/mL for OEE, HSE and HM and molarity for OB. The only possibility would be to transform the OB molarity into mg/mL but the concentrations are too different (see data tables) to put them together in a single figure. To facilitate the understanding of the results related to the ileum and colon longitudinal and circular smooth muscle spontaneous contractility, we have added two summary tables.

Line 390: It is unclear why OB is being utilized as a positive control in this scenario given it is noted in lines 386-387 that the effect of OB on cardiovascular parameters was assessed, suggesting the effect is currently unknown. Whereas a positive control is a treatment that is known to produce the expected effect to validate the experimental procedure. Since it seems as though OB is being treated as an intervention, was a control used for this assessment?

We apologize for the misunderstanding related to OB as "positive control" as included in the materials and methods paragraph. OB is the positive control for gastrointestinal applications. Given its biological action also due to the interaction with calcium channels, we considered it appropriate to study OB also on cardiovascular parameters considered in this context as "off target" in order to compare the biological actions of HM and individual extracts with those of OB.

Lines 400-411: This appears to be a description of methods; however, it is currently noted in the results section. Suggest revising the text to include this description in the methods section.

This appears to be a description of methods; however, it is currently noted in the results section. Suggest revising the text to include this description in the methods section.

(An additional minor note, L-citrulline is written as L-Cit later in line 408, yet this abbreviation was not previously indicated and does not seem necessary in this context.)

L-Cit was deleted in line 409 and substituted with “L-citrulline (reported in the Method section).

Line 420: Could this line please be clarified, specifically the determination and representation of the 12% and 35%?

We have removed this sentence and rewritten the whole description of the effects of the pure OEE on the eNOS activity. In the modified text we have reported that “the eNOS was inhibited, but only at the high doses of 5 and 10 mg/mL (97% and 98% inhibition, respectively). A residual eNOS inhibition of 12% was evaluated at 0.5 mg/mL, while a complete isoform selectivity (0% inhibition) was reached at the lowest dose of 0.1 mg/mL”. Moreover, we reported in Figure 10a and 10b the eNOS and iNOS inhibition percentages.

In order to reduce the repetition rate, we modified and summarized the description of the experimental part on NOS inhibition (lines 160-199), adding the appropriate reference. Nevertheless, details on the procedure followed to analyze the extracts were kept.

Lines 477-494: This section reads like content more applicable to a discussion section, suggest modifying accordingly.

Lines 518-521: Similar to a note above, it appears as though this results section is currently presenting a description of methodologies utilized as well as conjecture more relevant to a discussion section. Hence, this content, may be more beneficially presented in the respective methods and discussion sections.

We thank the Referee for his suggestion related to lines 477-494 and 518-521 and we fully agree that part of the content of these paragraphs is more suitable for the Discussion- rather than the Results-section. The text has been modified accordingly.

Lines 554-555: Could it please be clarified as to what “HM’s action trend is similar” to?

Sorry for the incomplete sentence. We have added the missing part.

Lines 594-597: This statement does not appear to be verified by the present findings, and there does not appear to be a reference to support this as it relates to HM, thus perhaps this should be presented as more of a possibility (using could or may phrasing) rather than a statement.

We agree with Referee that the above-mentioned sentence is misleading. Indeed, we were referring to a possibility but being this not sufficiently clear, the sentence has been remodulated as follows:

In order to reduce/avoid HM effects at cardiovascular level, a possibility could consist in the set- up of an appropriate tissue specific formulation able to selectively accumulate active compounds in the target organ (intestine), while keeping their concentration in off target organs (heart, blood vessels) below that which trigger cardiovascular effects. Interestingly, colon-specific drug delivery systems have been recently found, and these have had positive significant impact for the treatment of intestinal inflammation-related diseases [40, 45-47].

Lines 683-686: This paragraph appears to be a duplicate of the paragraph above. Please remove.

We remove the paragraph.

Thank-you for your time and consideration of these suggestions.

Round 2

Reviewer 1 Report

The Authors have made revisions according to my suggestions. I have no further comments.

Reviewer 2 Report

Thank-you for taking the time to make the modifications.